# Associations between community cultural engagement and life satisfaction, mental distress and mental health functioning using data from the UK Household Longitudinal Study (UKHLS): are associations moderated by area deprivation?

Hei Wan Mak,[1] Rory Coulter [ID],[2] Daisy Fancourt [ID][1]

¹Department of Behavioural Science and Health, University College London, London, UK
²Department of Geography, University College London, London, UK

**Correspondence to**
Dr Daisy Fancourt;
d.fancourt@ucl.ac.uk

## ABSTRACT

**Objectives** The association between community cultural engagement and mental health and well-being is well established. However, little is known about whether such associations are influenced by area characteristics. This study therefore examined whether the association between engagement in community cultural assets (attendance at cultural events, visiting museums and heritage sites) and subsequent well-being (life satisfaction, mental distress and mental health functioning) is moderated by neighbourhood deprivation.

**Design** Data were drawn from Understanding Society: The UK Household Longitudinal Study waves 2 and 5. Participating households' addresses were geocoded into statistical neighbourhood zones categorised according to their level of area deprivation.

**Setting** General population.

**Participants** UK general adult population, with a total sample of 14 783.

**Main outcome measures** Life satisfaction was measured with a seven-point scale (1: completely unsatisfied to 7: completely satisfied). Mental distress was measured using the General Health Questionnaire 12. Mental health functioning was measured using 12-item Short Form Health Survey (SF-12).

**Results** Using Ordinary Least Squares (OLS) regression, we found that engagement in cultural assets was consistently and positively associated with subsequent life satisfaction and mental health functioning and negatively associated with mental distress. Importantly, such associations were independent of individuals' demographic background, socioeconomic characteristics and regional location. The results also show that relationships between engagement in community cultural assets and well-being were stronger in more deprived areas.

**Conclusions** This study shows that engagement in community cultural assets is associated with better well-being, with some evidence that individuals in areas of high deprivation potentially may benefit more from these engagements. Given that causal mechanisms were

## STRENGTHS AND LIMITATIONS OF THIS STUDY

⇒ This is one of the first large, population-based study to investigate whether the association between community cultural engagement and mental well-being varies with local deprivation.

⇒ This study was based on a nationally representative sample with a 3-year follow-up period and the analysis controlled for a wide range of important confounding factors.

⇒ However, as this was an observational study, causality cannot be conclusively established.

⇒ The Index of Multiple Deprivation is composed of various weighted components (eg, living environment, income and employment deprivation), which are combined to create a proxy of neighbourhood deprivation.

not tested, causal claims cannot be generated from the results. However, the results suggest that place-based funding schemes that involve investment in areas of higher deprivation to improve engagement rates should be explored further to see if they can help promote better well-being among residents.

## INTRODUCTION

The role of community cultural engagement (CCE) in health and well-being has received increased attention across a number of disciplines over the past two decades. CCE can include going to cultural events (such as the theatre, opera, concerts or exhibitions) and visiting museums and heritage sites (including visiting a city or town with historic character) and has been associated with improvements in well-being,[1–3] slower declines in cognition,[4] reduced levels of isolation and loneliness[5], enhanced social

well-being in the community[6] and lower mortality rates.[7 8] These benefits have been found among healthy individuals as well as people with mental health problems, dementia, substance use addiction, carers, families living in deprived areas, asylum seekers and isolated adults.[9–11]

Unfortunately, previous studies have suggested a social and geographical gradient in cultural engagement, in which the engagement rate is higher among people living in more affluent areas.[12–14] Certainly, CCE relies on cultural assets (ie, tangible spaces, buildings or organisations) being available within communities, and most community cultural assets are unevenly distributed across the UK in ways that covary with deprivation.[15–17] In addition, even when cultural assets are available, the characteristics of neighbourhoods—such as the living environment, local facilities and services, social connections and networks, levels of accessibility and the safety of the area—have been identified as specific factors that can affect the engagement rate.[12 14 18 19] As such, some people may have greater, safer and easier opportunities to engage with culture (the 'contextual effect'). However, it is worth noting that this effect may be less obvious in those deprived parts of England that lie in close proximity to more affluent areas with greater availability of assets.[15–17]

It has also been shown that, independent of area deprivation, people living in particular areas tend to cluster together according to their demographics (eg, age and ethnicity), socioeconomic position (ability to afford housing) and lifestyle or cultural preferences. This can lead to the acquisition of common norms, values, and economic and cultural capital, which may lead to different behavioural patterns of CCE within specific neighbourhoods (known as a 'compositional effect').[20 21] Compositional effects have been demonstrated for CCE in previous studies. For instance, 'cosmopolitan' or 'student' areas of England have been identified as having particularly strong patterns of CCE relative to, say, postindustrial communities.[13]

However, variations in people's CCE behaviours may not be simply a matter of contextual and compositional effects. It is plausible that the place where people live is a *predictor* of CCE and a *moderator* of the association between CCE and health outcomes. There is evidence of similar social and geographical gradients in CCE and health, with many studies showing spatially variable associations between area deprivation and poor health outcomes, although results have been complex and were not entirely consistent.[22–24] Indeed, many studies examining health inequality have incorporated geographical data and have suggested that unequal access to health-promoting environments, such as the availability of and accessibility to cultural and artistic activities, could contribute to the spatial health divide.[12 25] This is particularly relevant to areas of deprivation where educational levels, employment rates and living standards are lower and mental health

problems are more common.[26] Therefore, it is plausible that bringing cultural resources to deprived areas and improving community infrastructure could have a greater impact on people's well-being in these areas than in affluent areas, as in deprived areas, there may otherwise be limited assets and opportunities to build positive mental health.

In light of this, it is relevant to explore whether place is important not just in predicting levels of CCE but also in moderating the relationship between CCE and health outcomes. Understanding whether there is any moderation is crucial and relevant to current public health strategies and interventions such as 'social prescribing' schemes and place-based funding streams for the cultural sector. These are predicated on the belief that increasing the local availability of assets and their usage could lead to increased CCE and thus improved health outcomes.[27–32] However, it remains unclear whether investment in cultural assets in different locations holds equal potential for positively influencing health.

Therefore, in this paper, we used a large longitudinal and nationally representative sample of adults to examine whether the association between engagement in community cultural assets and subsequent well-being (operationalised as life satisfaction, mental distress and mental health functioning) is moderated by geographical deprivation in the origin location. In particular, given that differential exposure to risks affects health differently, we explored whether individuals living in areas of high deprivation who are at higher risks of experiencing poorer mental health gain the same or even greater well-being benefits from CCE as individuals living in areas of low deprivation.

## DATA AND METHOD

We used data from Understanding Society: The UK Household Longitudinal Study (UKHLS), which is a continuation of the long-running British Household Panel Survey. UKHLS follows over 50 000 individuals from 30 000 households annually[33] and collects rich information on engagement in cultural events and museums and heritage. Crucially, the survey also collects a suite of measures of self-reported individual well-being.

In order to investigate the role of area deprivation, we used geo-coded UKHLS data in which participating households' addresses have been positioned into a number of spatial zoning systems (eg, administrative and census statistical geographies). For our analysis, we extracted a sample of adults living in England who responded to the wave 2 (2010/2012; response rate=84%), where data on engagement in CCE were first available, and wave 5 (2013/2015; response rate=85%) interviews. We only considered respondents who completed both waves 2 and 5 interviews and those who answered across all measures, as well as respondents with a valid sampling weight value.

## Patient and public involvement

This study used publicly available secondary data from the UK Data Service (https://www.ukdataservice.ac.uk/). Patients and the public were not involved.

## Measures

We defined neighbourhoods as 2011 census Lower Layer Super Output Areas (LSOA) and matched the 2011 LSOA with wave 2 UKHLS where data were collected between 2010 and 2012. LSOAs are designed for the consistent reporting of small area statistics in England and Wales. Using the LSOA geocodes, we attached the 2015 English Index of Multiple Deprivation (IMD),[15] which measures the relative deprivation of small areas across seven domains: income, employment, health deprivation and disability, education, skills and training, crime, barriers to housing and services and living environment. Our main analysis was based on the IMD decile rank values which we used as a continuous measure.

For CCE, we focused on attendance at cultural events, including going to the theatre, concerts, opera and exhibition and museums/galleries and heritage sites visits (a full list of CCE is provided in appendix I). CCE was measured in wave 2. At this wave, respondents were asked how often they had attended any of the cultural events, visited museums/galleries and visited heritage sites in the past 12 months. Frequency of engagement with these activities was categorised as 'not once in the last 12 months', 'once in the last 12 months', 'twice in the last 12 months', 'less often than once a month but at least 3 or 4 times a year', 'less often than once a week but at least once a month' and 'at least once a week'. Due to the similar nature of the activities, visits in museums/galleries and heritage sites were collapsed into one variable. Both types of CCE (cultural attendance and museum and heritage engagement) were treated as continuous measures.

We explored three outcome well-being measures in wave 5, which took place around 3 years after our wave 2 baseline. These measures were life satisfaction, mental distress and mental health functioning. Life satisfaction was measured through asking respondents how satisfied they felt with their life overall, with responses ranging from 1 (completely unsatisfied) to 7 (completely satisfied). This measure has been used as one of the questions to measure personal well-being in the UK general population by the UK Office for National Statistics, although the scale varies slightly.[34] Mental distress was measured using the General Health Questionnaire (GHQ-12), a screening device identifying psychiatric disorders in the general population and in primary medical care settings.[35 36] The GHQ-12 self-reported questionnaire includes 12 four-point items (such as sleeping problems, overall happiness and depressive symptoms; $\alpha=0.91$). The scale was computed additively, ranging from 1 to 4, with higher scores indicating a greater incidence of mental distress. Mental health functioning was measured using 12-item Short Form Health Survey (SF-12); $\alpha=0.90$), a well-validated survey that was designed to measure

respondents' general health-related quality of life. The 12-item survey contains eight indicators: physical functioning, role limitations due to physical health problems, bodily pain, general health, vitality, social functioning, role limitations due to emotional problems and mental health.[37] The scale was computed additively, ranging from 1 to 5, with higher scores indicating better levels of mental health functioning.

In our analysis, we controlled for broad regional variations (north, midlands and south) as well as demographic and socioeconomic variables in wave 2. Demographic control variables included age, gender (female vs male), ethnicity (white, Asian/Asian British, black/black British and mixed/others), partnership status (married/in cohabitation, single and never married/never in civil partnership and separated/divorced/widowed), presence of child(ren) under the age of 16 years and whether respondents were living alone. Socioeconomic controls included educational level (university degree, advanced (A-levels) exams/higher education (eg, a Higher Education Certificate/Business and Technology Education Council (BTEC)), passed General Certificate of Secondary Education (GCSE) or equivalent and unrecognised qualification/no qualification), occupational socioeconomic status (managerial/professional, intermediate/small employment/own account, lower supervision/lower technician/semi-routine/routine, and not in employment (eg, unemployed or retired or students)), household monthly gross income (logged), and housing tenure (house owner, social rent and private rent).

## Statistical analysis

To understand whether the relationship between CCE (x) and mental well-being measures (y) varied with area deprivation (the potential moderator), we ran a cross-sectional analysis using Ordinary Least Squares (OLS) regression models. Given that residential location is highly correlated with personal demographic and socioeconomic factors, the regression models were constructed sequentially to understand the changes of the association between CCE and mental well-being. In model 1 (basic model), we included only CCE. Model 2 additionally controlled for Index of Multiple Deprivation (IMD) and the interaction terms (ie, CCE*IMD). Model 3 additionally included demographic factors, and finally model 4 adjusted for socioeconomic position. All models were weighted using inverse probability weights derived from the wave 2 longitudinal weights supplied with UKHLS. These weights have been tailored to the analytical sample and should correct our estimates to take into account differential sample selection and retention probabilities.[38] List-wise deletion was used to handle small volumes of missing data (0.3%).

To check whether our data met the assumption of OLS regressions, we ran a series of regression diagnostic tests. Our tests show that the distribution of residuals was almost homoscedastic and normal for models estimating mental distress and mental health functioning. The

distribution was less satisfactory for models estimating life satisfaction, most likely due to the more discrete scale of this variable. However, our large sample size and the fact we present weighted estimates with robust standard errors should mitigate against problems arising from model fit. Nonetheless, more caution should be exercised when interpreting the results of the life satisfaction models as compared with the other well-being outcomes. The risk of multicollinearity was also very low with a mean variance inflation factor (VIF) of 2.03 among the independent variables.

As further robustness checks, all analyses were replicated with two threefold deprivation measures using 20% IMD threshold and 10% IMD threshold (ie, 10%/20% most deprived areas, 10%/20% least deprived areas and the remaining intermediate neighbourhoods). All analyses were carried out using Stata V.16.

## RESULTS

In our sample, 38 069 participants living in England responded to the wave 2 and 30 635 participants responded to the wave 5 interviews. A total 25 464 individuals completed both interviews. Of these, 23 247 participants provided answers for the outcome variables, with 23 244 respondents additionally answered questions on CCE, and 22 463 individuals also answered across all other measures. Among them, 14 833 received a non-zero wave 2 longitudinal weight value provided by UKHLS. Of these, all but 50 (14 783) received a valid tailored weight value for this analysis (table 1).

Table 1 reports weighted descriptive statistics of the unweighted and weighted samples. In our weighted sample, the average age was 47 years, 52% were female and over 90% were of white ethnicity. In relation to socioeconomic position, 35% of the respondents had a university degree, 27% were in managerial/professional roles and 72% owned a house. On average, the scores of life satisfaction, mental distress and mental health functioning were 5.1, 1.9 and 3.8, respectively. Twenty-two per cent reported of not attending any cultural events and 27% reported of not visiting museums or heritage sites in the past 12 months. Around 19% attended cultural events and 15% visited museums or heritage sites at least once a month. (tables 2–4)

## Life satisfaction

After adjusting for individual demographic factors and socioeconomic position, both types of CCE were associated with a slightly greater level of life satisfaction (attending cultural events: coef=0.08, 95% CI 0.04 to 0.13, standardised beta=0.08; visiting museums and heritage sites: coef=0.06, 95% CI 0.02 to 0.11, beta=0.06) (table 2). There was no evidence that this association was moderated by area deprivation when using the continuous measure, nor when using the threefold 10% or 20% measure (table 2, online supplemental tables S1 and S2).

**Table 1** Descriptive statistics of the unweighted and weighted samples

| | Unweighted (n=22 463) | Weighted (n=14 783) |
|---|---|---|
| | Proportion/ mean (SE) | Proportion/ mean (SE) |
| Community cultural engagement, wave 2 | | |
| Cultural events* | 2.19 (1.47) | 2.22 (0.01) |
| Museums and heritage sites* | 1.95 (1.50) | 1.96 (0.01) |
| Mental well-being, wave 5 | | |
| Life satisfaction† | 5.03 (1.51) | 5.06 (0.01) |
| Mental distress (GHQ-12, ranging from 1 to 4) | 1.93 (0.00) | 1.92 (0.00) |
| Mental health functioning (SF-12; ranging from 1 to 5) | 3.74 (0.00) | 3.76 (0.01) |
| Demographic backgrounds, wave 2 | | |
| Age | 47.8 (16.9) | 46.9 (0.17) |
| Gender (%) | | |
| Female | 56.7 | 51.8 |
| Male | 43.3 | 48.2 |
| Ethnicity (%) | | |
| White | 86.1 | 90.7 |
| Asian/Asian British | 7.89 | 5.37 |
| Black/black British | 3.66 | 2.22 |
| Mixed/other | 2.35 | 1.76 |
| Living alone (%) | | |
| No | 84.8 | 84.8 |
| Yes | 15.2 | 15.2 |
| Partnership status (%) | | |
| Single and never married | 17.7 | 21.9 |
| Married or in cohabitation | 67.8 | 64.3 |
| Separated or divorced or widowed | 14.5 | 13.8 |
| Responsible for child(ren) under 16 years (%) | | |
| No | 80.8% | 83.6% |
| Yes | 19.3% | 16.4% |
| Regions (%) | | |
| North (North East, North West and Yorkshire and the Humber) | 27.9 | 28.2 |
| Midlands (East Midlands and West Midlands) | 19.9 | 18.8 |
| South (London, South East, South West and East) | 52.2 | 53.1 |
| Socioeconomic position, wave 2 | | |
| Educational levels (%) | | |
| University degree | 37.2 | 35.2 |

Continued

**Table 1** Continued

| | Unweighted (n=22 463) | Weighted (n=14 783) |
|---|---|---|
| | Proportion/ mean (SE) | Proportion/ mean (SE) |
| Advanced (higher education/ A-level) | 19.6 | 20.2 |
| GCSE or equivalent | 21.4 | 21.3 |
| Unrecognised/no qualification | 21.9 | 23.3 |
| Occupational socioeconomic status (%) | | |
| Managerial/professional | 28.5 | 27.4 |
| Intermediate/small employment/own account | 15.7 | 15.5 |
| Lower supervision/lower technician/semi-routine/ routine | 21.0 | 22.5 |
| Not in employment (incl. retired, full-time student) | 34.7 | 34.3 |
| Household monthly gross income (%) | | |
| £0–£1015 | 21.1 | 21.4 |
| £1015–£1551 | 24.2 | 24.4 |
| £1551–£2355 | 26.4 | 26.1 |
| £2355–£32 622 | 28.3 | 28.2 |
| Housing tenure (%) | | |
| House owner | 74.2 | 72.1 |
| Social rent | 15.4 | 15.9 |
| Private rent | 10.5 | 12.0 |
| Levels of area deprivation‡ | 5.66 (2.84) | 5.70 (0.03) |

*A six-point scale, ranging from 'not once in the last 12 months', 'once in the last 12 months', 'twice in the last 12 months', 'less often than once a month but at least three or four times a year', 'less often than once a week but at least once a month' to 'at least once a week'.
†Life satisfaction was measured using a scale from 1 'completely unsatisfied' to 7 'completely satisfied'.
‡Levels of area deprivation was derived from the Index of Multiple Deprivation, which has a scale from 1: 'most deprived 10%' to 10: 'least deprived 10%'.
GCSE, General Certificate of Secondary Education; GHQ-12, 12-item General Health Questionnaire; SF-12, 12-item Short Form Health Survey.

### Mental distress (GHQ-12)

When accounting for individual demographic and socioeconomic characteristics, attending cultural events was associated with lower levels of mental distress (coef=−0.03, 95% CI=−0.04 to −0.01, beta=−0.09) and so was visiting museums and heritage sites (coef=−0.02, 95% CI −0.03 to −0.00, beta=−0.05). There was some evidence of moderation by area deprivation shown in cultural attendance, although the moderation was attenuated when adjusting for socioeconomic position (table 3). To explore the interaction effects, we estimated marginal effects. When comparing with the most distinct cultural attendance

frequency (ie, at least once a week vs none in the past 12 months), the score of mental distress among people living in the 10% least deprived areas was decreased from 1.89 (no engagement) to 1.83 (weekly engagement). Similarly, of people living in the 10% most deprived areas, their mental distress score was decreased from 2.05 (no engagement) to 1.89 (weekly engagement). The differences in mental distress scores between people living in varying levels of area deprivation became smaller with increased cultural attendance frequency (figure 1). No moderation association was found for museum and heritage engagement, nor when using the threefold 20% IMD measure (online supplemental table S3) and threefold 10% IMD measure (online supplemental table S4) to test for non-linear moderations.

### Mental health functioning (SF-12)

After adjusting for demographic background and socioeconomic position, both types of CCE were associated with a higher level of subsequent mental health functioning (attending cultural events: coef=0.06, 95% CI 0.04, 0.08, beta=0.13; visiting museums and heritage sites: coef=0.05, 95% CI 0.03 to 0.06, beta=0.10). There was some indication of moderation of the association by area deprivation for both types of CCE, although the moderation was attenuated when adjusting for socioeconomic position for cultural engagement and when adjusting for demographic factors for museum and heritage engagement (table 4). Of those living in the 10% least deprived areas, their score in mental health functioning was increased from 3.79 to 3.80 (no engagement) to 4.07 (weekly engagement) for both types of CCE. Among those living in the 10% most deprived areas, their score was increased from 3.39–3.44 (no engagement) to 3.85–3.86 (weekly engagement). The differences in mental health functioning scores between people living in varying levels of area deprivation became narrower with increased CCE (figures 2 and 3). The moderation associations were also reflected when using the threefold IMD measures to test for non-linearities, although they were less prominent with the 20% threshold (online supplemental tables S5 and S6).

### Sensitivity analyses

Additional sensitivity checks were conducted using outcome data from wave 2 rather than wave 5. In these checks, we estimated the same models as discussed above but this time relating wave 2 CCE and IMD values to contemporaneously measured wave 2 life satisfaction, mental distress and mental health functioning outcomes. The results were broadly in line with the findings discussed above and are available from the lead author on request.

### DISCUSSION

This is one of the first large population-based study to investigate whether the association between CCE (attending cultural events or visiting museums and heritages sites)

**Table 2** OLS regression estimating the association between community cultural engagement (CCE) at wave 2 and life satisfaction at wave 5 (weighted; n=14783)

| | M1: basic model | | | M2: M1 +IMD and interaction terms | | | M3: M2 +demographic factors | | | M4: M3 +socioeconomic position | | |
|---|---|---|---|---|---|---|---|---|---|---|---|---|
| | Coef | 95% CI | P value | Coef | 95% CI | P value | Coef | 95% CI | P value | Coef | 95% CI | P value |
| Cultural events | **0.10** | **0.08 to 0.12** | **0.000** | **0.10** | **0.05 to 0.15** | **0.000** | **0.12** | **0.07 to 0.16** | **0.000** | **0.08** | **0.04 to 0.13** | **0.000** |
| IMD | | | | **0.07** | **0.05 to 0.09** | **0.000** | **0.06** | **0.04 to 0.08** | **0.000** | **0.04** | **0.02 to 0.06** | **0.000** |
| Cultural events*IMD | | | | −0.00 | −0.01 to 0.00 | 0.353 | −0.00 | −0.01 to 0.00 | 0.300 | −0.00 | −0.01 to 0.00 | 0.516 |
| Constant | **4.84** | **4.79 to 4.89** | **0.000** | **4.47** | **4.36 to 4.59** | **0.000** | **4.35** | **4.18 to 4.52** | **0.000** | **4.08** | **3.68 to 4.47** | **0.000** |
| R-squared | 0.01 | | | 0.02 | | | 0.03 | | | 0.04 | | |
| Museums and heritage sites | **0.11** | **0.09 to 0.13** | **0.000** | **0.11** | **0.06 to 0.15** | **0.000** | **0.10** | **0.05 to 0.14** | **0.000** | **0.06** | **0.02 to 0.11** | **0.006** |
| IMD | | | | **0.07** | **0.05 to 0.09** | **0.000** | **0.06** | **0.05 to 0.08** | **0.000** | **0.04** | **0.02 to 0.06** | **0.000** |
| Museums and heritage sites*IMD | | | | −0.00 | −0.01 to 0.00 | 0.235 | −0.00 | −0.01 to 0.00 | 0.267 | −0.00 | −0.01 to 0.00 | 0.618 |
| Constant | **4.85** | **4.80 to 4.90** | **0.000** | **4.50** | **4.40 to 4.61** | **0.000** | **4.49** | **4.33 to 4.65** | **0.000** | **4.16** | **3.77 to 4.54** | **0.000** |
| R-squared | 0.01 | | | 0.02 | | | 0.03 | | | 0.04 | | |

M1 (basic model) included CCE. M2 additionally controlled for Index of Multiple Deprivation (IMD) and interaction terms (CCE*IMD). M3 additionally adjusted for demographic factors (age, gender, ethnicity, partnership status, presence of children under age 16 years, whether or not living alone and regional locations). M4 additionally adjusted for socioeconomic position (educational levels, occupational socioeconomic status, household monthly gross income and housing tenure). Bold values denote statistical significance at the p<0.05 level.

**Table 3** OLS regression estimating the association between community cultural engagement (CCE) at wave 2 and mental distress (GHQ-12) at wave 5 (weighted; n=14783)

| | M1: basic model | | | M2: M1 +IMD and interaction terms | | | M3: M2 +demographic factors | | | M4: M3 +socio-economic position | | |
|---|---|---|---|---|---|---|---|---|---|---|---|---|
| | Coef | 95% CI | P value | Coef | 95% CI | P value | Coef | 95% CI | P value | Coef | 95% CI | P value |
| Cultural events | **-0.02** | **-0.03 to -0.02** | **0.000** | **-0.03** | **-0.05 to -0.02** | **0.000** | **-0.04** | **-0.05 to -0.02** | **0.000** | **-0.03** | **-0.04 to -0.01** | **0.000** |
| IMD | | | | **-0.02** | **-0.03 to -0.01** | **0.000** | **-0.02** | **-0.02 to -0.01** | **0.000** | **-0.01** | **-0.02 to -0.01** | **0.000** |
| Cultural events*IMD | | | | **0.00** | **0.00 to 0.00** | **0.012** | **0.00** | **0.00 to 0.00** | **0.033** | 0.00 | -0.00 - 0.00 | 0.107 |
| Constant | **1.97** | **1.96 to 1.99** | **0.000** | **2.07** | **2.04 to 2.11** | **0.000** | **2.11** | **2.05 to 2.16** | **0.000** | **2.21** | **2.09 to 2.32** | **0.000** |
| R-squared | 0.01 | | | 0.01 | | | 0.03 | | | 0.04 | | |
| Museums and heritage sites | **-0.02** | **-0.03 to -0.02** | **0.000** | **-0.03** | **-0.04 to -0.01** | **0.000** | **-0.03** | **-0.04 to -0.01** | **0.000** | **-0.02** | **-0.03 to -0.00** | **0.020** |
| IMD | | | | **-0.02** | **-0.02 to -0.01** | **0.000** | **-0.02** | **-0.02 to -0.01** | **0.000** | **-0.01** | **-0.02 to -0.01** | **0.000** |
| Museums and heritage sites*IMD | | | | 0.00 | -0.00 to 0.00 | 0.106 | 0.00 | -0.00 to 0.00 | 0.101 | 0.00 | -0.00 to 0.00 | 0.425 |
| Constant | **1.97** | **1.95 to 1.98** | **0.000** | **2.05** | **2.02 to 2.08** | **0.000** | **2.06** | **2.01 to 2.11** | **0.000** | **2.17** | **2.06 to 2.29** | **0.000** |
| R-squared | 0.01 | | | 0.01 | | | 0.03 | | | 0.04 | | |

M1 (basic model) included CCE. M2 additionally controlled for Index of Multiple Deprivation (IMD) and interaction terms (CCE*IMD). M3 additionally adjusted for demographic factors (age, gender, ethnicity, partnership status, presence of children under age 16 years, whether or not living alone and regional locations). M4 additionally adjusted for socioeconomic position (educational levels, occupational socio-economic status, household monthly gross income and housing tenure). Bold values denote statistical significance at the p<0.05 level. GHQ-12, 12-item General Health Questionnaire.

**Table 4** OLS regression estimating the association between community cultural engagement (CCE) at wave 2 and mental health functioning (SF-12) at wave 5 (weighted; n=14783)

| | M1: basic model | | | M2: M1 +IMD and interaction terms | | | M3: M2 +demographic factors | | | M4: M3 +Socio-economic position | | |
|---|---|---|---|---|---|---|---|---|---|---|---|---|
| | Coef | 95% CI | P value | Coef | 95% CI | P value | Coef | 95%CI | P value | Coef | 95%CI | p-value |
| Cultural events | 0.10 | 0.10 to 0.11 | 0.000 | 0.12 | 0.10 to 0.14 | 0.000 | 0.10 | 0.08 to 0.12 | 0.000 | 0.06 | 0.04 to 0.08 | 0.000 |
| IMD | | | | 0.05 | 0.04 to 0.05 | 0.000 | 0.05 | 0.04 to 0.05 | 0.000 | 0.03 | 0.02 to 0.04 | 0.000 |
| Cultural events*IMD | | | | −0.01 | −0.01 to −0.00 | 0.000 | −0.00 | −0.01 to −0.00 | 0.008 | −0.00 | −0.00 - 0.00 | 0.183 |
| Constant | 3.53 | 3.50 to 3.55 | 0.000 | 3.30 | 3.25 to 3.35 | 0.000 | 3.76 | 3.68 to 3.83 | 0.000 | 3.68 | 3.51 to 3.84 | 0.000 |
| R-squared | 0.05 | | | 0.07 | | | 0.12 | | | 0.16 | | |
| | | | | | | | | | | | | |
| Museums and heritage sites | 0.08 | 0.07 to 0.09 | 0.000 | 0.09 | 0.07 to 0.11 | 0.000 | 0.08 | 0.06 to 0.10 | 0.000 | 0.05 | 0.03 to 0.06 | 0.000 |
| IMD | | | | 0.04 | 0.03 to 0.05 | 0.000 | 0.04 | 0.03 to 0.05 | 0.000 | 0.02 | 0.02 to 0.03 | 0.000 |
| Museums and heritage sites*IMD | | | | −0.00 | −0.01 to −0.00 | 0.027 | −0.00 | −0.01 to 0.00 | 0.086 | 0.00 | −0.00 to 0.00 | 0.948 |
| Constant | 3.60 | 3.57 to 3.62 | 0.000 | 3.40 | 3.35 to 3.45 | 0.000 | 3.87 | 3.80 to 3.94 | 0.000 | 3.74 | 3.58 to 3.90 | 0.000 |
| R-squared | 0.03 | | | 0.05 | | | 0.12 | | | 0.16 | | |

M1 (basic model) included CCE. M2 additionally controlled for Index of Multiple Deprivation (IMD) and interaction terms (CCE*IMD). M3 additionally adjusted for demographic factors (age, gender, ethnicity, partnership status, presence of children under age 16 years, whether or not living alone and regional locations). M4 additionally adjusted for socioeconomic position (educational levels, occupational socioeconomic status, household monthly gross income and housing tenure). Bold values denote statistical significance at the p<0.05 level.

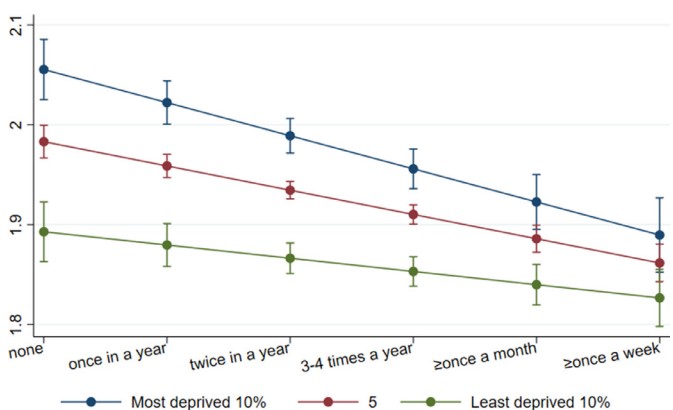

**Figure 1** Association between cultural attendance and mental distress by levels of area deprivation. Data: UKHLS, waves 2 and 5. Estimates are derived from an OLS regression model including an interaction term (cultural attendance*IMD). IMD, Index of Multiple Deprivation.

and mental well-being (including life satisfaction, mental distress and mental health functioning) varies with local deprivation. In line with previous research, our results show that engagement in cultural assets was consistently and positively associated with subsequent life satisfaction and mental health functioning and negatively associated with mental distress.[3] Importantly, this paper found that such associations were independent of individuals' demographic background, socioeconomic characteristics and regional locations. In particular, our models show that every one SD increase in CCE is associated with higher life satisfaction and mental health functioning (by 0.06–0.13 SD) and lower mental distress (by 0.05–0.09 SD). Although the magnitude of these effects is fairly small, such associations were evident even after considering levels of area deprivation, demographics and socioeconomic factors and when predicting the outcomes measured after 3 years. This suggests that, while social and geographical factors can influence engagement rates,[12–14 18 39] CCE is consistently associated with minor

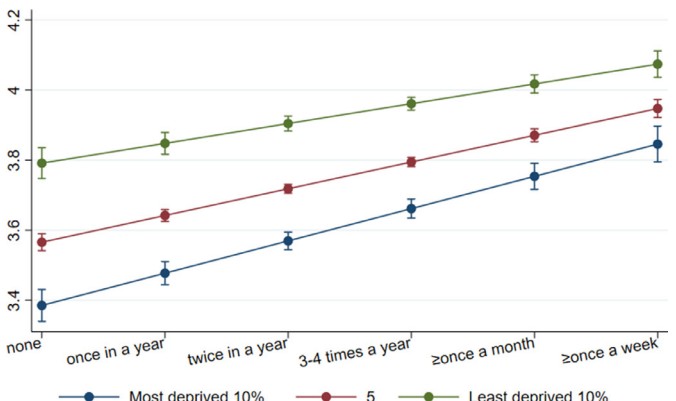

**Figure 2** Association between cultural attendance and mental health functioning by levels of area deprivation. Data: UKHLS, waves 2 and 5. Estimates are derived from an OLS regression model including an interaction term (cultural attendance*IMD). IMD, Index of Multiple Deprivation.

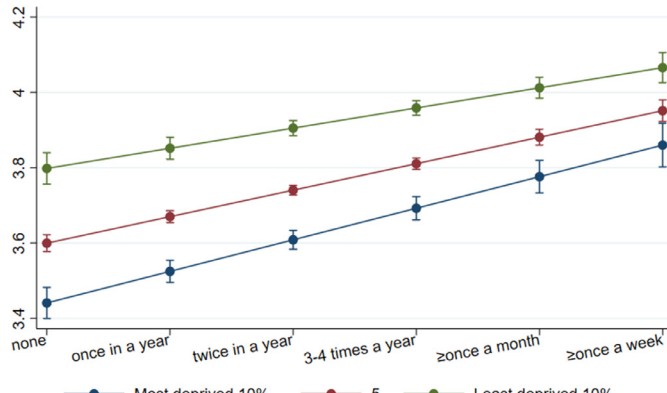

**Figure 3** Association between museum and heritage engagement and mental health functioning by levels of area deprivation. data: UKHLS, waves 2 and 5. Estimates are derived from an OLS regression model including an interaction term (museum and heritage engagement*IMD). IMD, Index of Multiple Deprivation.

improvements in mental well-being regardless of where people live.

Notably, our analysis shows some indications of interactions between both types of CCE and area deprivation. In particular, we found that the rates of growth in mental health functioning that accompany CCE are stronger among people living in deprived areas and that the rates of decline in mental distress that accompany cultural events attendance are also more prominent among those living in deprived areas. This suggests that individuals who live in highly deprived areas may benefit the most from CCE in terms of mental well-being improvements. This provides an important insight by suggesting that when individuals living in deprived areas are offered the opportunities to engaging in CCE, they could potentially experience greater improvements in mental well-being than those living in wealthy areas who have already been able to benefit from higher levels of engagement (possibly since childhood) and who usually enjoy other social advantages (eg, higher social positions) that fortify their mental well-being. In addition, the differences in mental well-being between people living in varying levels of deprivation become somewhat smaller as CCE increases. Despite this, deprivation appears to be a consistent barrier to engagement.[13 14 18] People living in deprived areas are less likely to engage in CCE not simply because of their personal background and characteristics (eg, lower educational attainment and occupational position), but because of the areas they reside in offer less cultural opportunities (eg, unsafe, culturally deprived and undesirable).[14 18] Consequently, unequal access to the arts may inhibit people living in deprived communities from enjoying the benefits provided by CCE and hence exacerbate social, cultural and health inequalities. No moderations are found for life satisfaction, nor for engagement in museum and heritage and mental distress. This suggests that the benefits of CCE on life satisfaction and mental distress are similar regardless of residential locations.

The results of this study have clear implications for the design and roll-out of place-based programmes for CCE that operate under the assumption that investing in areas of high deprivation and low cultural opportunity could improve well-being levels. One example of such a scheme is the UK 'Creative People and Places' funded by Arts Council England. The data presented here suggest that such schemes could have the potential to achieve their aims as there is some indication that the benefits of CCE may be slightly more pronounced in more deprived areas. However, there are mixed implications for other schemes that aim to signpost individuals to existing community cultural activities to improve health and well-being, as occurs in 'social prescribing'. While such an approach clearly has some potential to achieve its health aims in the same way as place-based funding schemes, social prescribing schemes rely on the availability of community cultural assets within communities. It has been suggested that CCE can be increased in areas with more cultural opportunity structures and in areas with high accessibility to cultural infrastructures, especially for groups that have often been excluded from CCE.[12] The success of such schemes will therefore rely on their ability to overcome the psychological, logistical or structural barriers that could limit individuals' engagement within more deprived areas.

A key strength of this paper is that it is based on a nationally representative sample, with a 3-year follow-up period. Our analyses controlled for a wide range of potentially confounding factors and were largely consistent across three different measures of area deprivation. However, our study is not without its limitations. First, causality cannot be established given that this is an observational study where other unobserved factors may still confound the association between CCE and mental well-being. The relatively low levels of model fit and fairly small effect sizes make it worthwhile for further research to test how additional sociodemographic measures not routinely collected by social surveys might shape well-being outcomes. Despite this, a number of previous studies have already confirmed that CCE can be causally linked to well-being through randomised interventions,[1 2] and the generalisability of such findings has been suggested through causal inference analyses of longitudinal data in other papers.[3]

As another limitation, the Index of Multiple Deprivation is composed of various components (eg, living environment, income and employment deprivation) as a proxy of neighbourhood effect. However, it will be important to understand from future studies how other neighbourhood characteristics (eg, demographic structure and population density) may also influence the engagement level. Relatedly, future study may be needed to replicate the research on young people's data, given that the moderating effects of area deprivation may be more noticeable during childhood when young people are less able to travel to engage in cultural activities outside of their immediate residential area. Finally, due to data limitations, we were unable to identify whether CCE was recent or more long-standing. It is likely that people who engage in culture and heritage over long periods of time may experience a greater level of mental well-being. Future research may also want to explore other mental health measures that are more commonly used in clinical practices, such as the Patient Health Questionnaire and General Anxiety Disorder measure.

## CONCLUSION

Our study provides insights into the role of place in long-term relationships between engagement in community cultural assets (both cultural events and museums and heritage sites) and mental well-being. Specifically, a higher CCE rate is associated with somewhat greater levels of life satisfaction and mental health functioning and with reduced mental distress. Notably, such relationships are independent of individual demographic and socioeconomic characteristics, and CCE is consistently associated with higher well-being regardless of where people live. As such, individuals living in areas of high deprivation can experience the same well-being benefits from CCE as individuals living in areas of low deprivation. Furthermore, there is also some evidence of moderation, with individuals in areas of high deprivation potentially even able to benefit more from CCE in terms of mental health functioning and improvements in mental distress. However, this does not mitigate the problem that individuals in areas of high deprivation are less likely to engage in CCE. This, therefore, suggests the importance of exploring further the effects of place-based funding schemes that involve investment in areas of higher deprivation to improve engagement rates to confirm if such schemes could help to promote higher levels of well-being among individuals in such areas. Findings from this study warrant further research in other datasets.

**Contributors** HWM conducted the data management, data analyses and provided input on the manuscript. RC and DF assisted with analytical issues and provided input on the analytical scheme and the manuscript. All authors are responsible for reported research and have participated in the concept and design, analysis and interpretation of data, and drafting and revising of the manuscript.

**Funding** DF is supported by the Wellcome Trust (205407/Z/16/Z). HWM is funded through the AHRC project HEARTS (AH/P005888/1). This project is also supported by UCL Grand Challenges Award 'Environment and Well-being' (551987/100/15642533002) and ESRC WELLCOMM project (ES/T006994/1).

**Competing interests** None declared.

**Patient consent for publication** Not required.

**Ethics approval** The University of Essex Ethics Committee has approved all data collection on Understanding Society: The UK Household Longitudinal Study main study and innovation panel waves, including asking consent for all data linkages except to health records. Respondents aged 16 years or above provided written consent to participate.

**Provenance and peer review** Not commissioned; externally peer reviewed.

**Data availability statement** Data may be obtained from a third party and are not publicly available. UKHLS data is available from the UK Data Service https://discover.ukdataservice.ac.uk/catalogue/?sn=6614. Data documentation is available from the Understanding Society website https://www.understandingsociety.ac.uk/documentation. Deprivation data are captured through linking Lower Layer Super

Output areas (LSOA) data from the UKHLS with data from the English Index of Multiple Deprivation (IMD 2015).

**ORCID iDs**
Rory Coulter http://orcid.org/0000-0002-0773-8919
Daisy Fancourt http://orcid.org/0000-0002-6952-334X

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
