## [Reviewer comments · BMJ Open]

ARTICLE DETAILS

TITLE (PROVISIONAL)	Associations between community cultural engagement and life satisfaction, mental distress and mental health functioning using data from the UK Household Longitudinal Study (UKHLS): are associations moderated by area deprivation?
AUTHORS	Mak, Hei Wan; Coulter, Rory; Fancourt, Daisy

VERSION 1 – REVIEW

REVIEWER	Hemingway, Ann Bournemouth University
REVIEW RETURNED	18-Jan-2021

GENERAL COMMENTS	Thank you for this very interesting and important paper, I just have some minor concerns regarding the title, and the measures analysed. You mention wellbeing in your title and then mention mental wellbeing in your abstract. The measures you have focused your analysis on are life satisfaction, mental distress and mental health functioning. It may be more appropriate to either clarify your focus in your title or build some links in the paper between these measures and overall definitions of wellbeing. Overall a really interesting and relevant paper thank you.
--

REVIEWER	Westerholm, Peter Uppsala University, Medical Sciences - Occupational and Environmental Medicine
REVIEW RETURNED	01-Feb-2021

GENERAL COMMENTS	The authors have picked on an interesting subject to inform colleagues of Medical Professions on the links between involving health professionals on community cultural assets and programmes and subsequent wellbeing and lifelong life satisfaction of subjects / citizens taking visiting museums and heritage sites into account and also qualitative conditions of neighbourhood deprivation as observed by the authors. The document submitted to The BMJ Open aims to describe a study addressing the possible association between engagement of a study population in community cultural assets such as attendance at cultural events, visiting museums and heritage sites and subsequent well-being (life satisfaction, mental distress and mental health functioning exploring the question whether such effects may be moderated by area neighbourhood deprivation. The study population of this undertaking is a UK general adult population with a total sample of 17, 948 subjects. It is based on a 3- year study based on Using an OLS regression analytic technique the authors observed that positive engagement of study population resulted in improvements of mental wellbeing and
---

	subsequent life satisfaction. At the same time a negative association was observed of such cultural association and health endpoint Mental distress in the study population. Basically three well-being indicators are to be explored – Life satisfaction, mental distress and mental health functioning. The authoring group has at the same time observed supportive evidence for study populations recruited from deprived areas – implying a combination of multiple Derivation components with weakness of combined components of Living Environment , Income and Employment Deprivation, used as a proxy of neighbourhood impact. So, in authors own assessment this study shows that engagements in cultural community assets are associated with improvement of well-being of population studied. The subject of deprivation of populations or areas involved in framing of research question(s) is certainly an innovative aspect in projects of this type. There is only scarce documentation available on this subject. The manuscript is based on data drawn from the database of the UK Household Longitudinal Study (UKHLS) community. Reviewers´overall assessment of manuscript scientific relevance and qualities is favourable in observing its demonstration of engagement in cultural aspects in a study population to be positively associated with improvement of life satisfaction and perceptions of mental health functioning and , conversely, negatively associated with signs indicating mental distress. To reviewer the manuscript as presently drafted opens up an aspect of editing contents to be attractive to readers of BMJ Open access Research Journal. The ideal solution to me to edit draft texts to offer clearly distinctive components of “Introduction”, “Background” moving over to “Research questions/ Issues” and from there on to “Study Population(s)”, “Information sources” and an overview of “Methods” describing the mechanics of how it was all managed including a time schedule of management of study as actually prepared and performed deserves a concentrated effort The section “Statistical Methods” is an obvious prelude to presenting “Results”. It is certainly not my preference to be identified as orthodox in editing of scientific draft manuscripts for publishing. I did, however, observe that my first impression when reading submitted manuscript was to react in seeing need of an experienced hard-nosed editors hand to go through text in making it effective and attractive to readers, in avoiding repetitions and redundancies in explanations. My spontaneous comments/questions follow here:  - 1. Abstract. The abstract is to impeccable in its concentration on Life satisfaction, mental health function possible.ning and mental stress and giving some selected results. - 2. Commendable emphasis of study being based on nationally representative population with 3-year FU period - 3. Necessary to claim causality as not being possible to establish? Reviewer: Yes – but it may still be suggestive - 4. Important Confounding factors controlled Is this Correct ? Which CF:s have been examined and assessed by authors? - 5. Ethics: Was improved consent by study population ascertained at any stage ? How was this done ? Which were the questions asked ? 6. Tables 1 – 2 – 3 - 4 All these tables require explanatory texts for benefit of readers. Where is this information to be included in manuscript for
--	---

	conclusive review ? For proper understanding of authors intention regarding association between community cultural engagement (CCE) and mental health distress (GHQ-12) at wave 5 of authors intentions and implications in reference made to “OLS regression estimating the association “ For tables 2-3-4 there is a need to explain “notes on IMD (ref: medium) 20% deprived 20% least deprived.....etc. For the benefit of readers of eventually published document. For figures 1 and 2 there is in the submitted draft manuscript package a draft said to be a figure legend explaining figures. I have not found it in manuscript under review 7 I see that the references to bibliography are assembled as a plain listing of authors names , documents referred to and bibliographical details. Is this in pursuance of BMJ Open practices ? Conclusive reviewers assessment:  - This is an interesting manuscript. It deserves to be taken seriously. Its main and principal value lies in an aspect of originality in observing findings supportive of suggesting area of study population (residential area or work area ?) to be capable of “moderating” effects of communal cultural engagement CEE, for instance with regard to end-point variables mental health functioning and prevalence of mental distress. This observation is at the present time of suggestive nature. - A challenging observation of authors is also the somewhat paradoxical observation in context of the study that subjects residing in areas of high deprivation may benefit more from such engagement, opening up a discussion of validity of place-based investment schemes in areas of high deprivation. - This observation needs to be qualified and verified in other studies to be regarded as established, reproducible and valid in assessing strategies of place-based schemes aiming at health promotion - The manuscript needs a hard-nosed Editors editing examination to go through text as presently drafted including texts of tables and figures. - Reviewers recommendation is to include a separate statistical review of analyses and methods. - The main study is based on following up of a study population. It is therefore necessary to tell readers about how this was done and its completeness. Were there losses to the follow-up? Have the losses to Follow-ups of the study population affected conclusions or interpretation of authors findings ? I am here adding a general comment on the submitted manuscript and the tables presenting the material and the analyses. 1. General comment on tables 1 through 4 The contents of all four tables consist of presentations of findings and observations characterizing study populations. Imputation of material to be complementary has apparently been carried out. Details on this procedure explaining what and how this was actually done not clarified in manuscript as presently drafted.
--	--

	In table 2 OLS regression analyses are presented on CCE and life satisfaction weighted at wave 5. Observations given of regression as interaction terms, socioeconomic positions and demographic terms. Under rubric “statistical analyses” reference is made to confounding factors and their construction based on their use in predictive models of analysis. Which confounding variables were taken into account ? How was this control actually implemented ? Under rubric ”Strengths and limitations of this study” is stated that in this observational study “causality can not be established”. In reviewer’s view this general statement on possibility to establish causality in a statistical association seen in an observational study seems me to be redundant. Is it really needed? The issue in itself may indeed deserve a comment. We know the existing differences of opinion on this issue in our own professional circles. On grounds of evidence in this manuscript and also in other studies evidence of suggestive nature may well be construed. As to acceptability of manuscript as presently drafted for BMJ Open publishing I recommend to have it examined in a statistical review of statistical methods and techniques used, in techniques used in supplementing study population with imputed subjects and groups and also in drafting of explanatory notes and comments to tables presenting results. Reference made to factors confounding observed associations of engagement in community cultural assets deserve special attention. This manuscript now reviewed has been approved in UK HOUSHOLD (UKHLS) in addressing main study, innovation panel waves and informed consent for all data linkages excepting health records. This is an excellent move. Reviewers question: How was this informed consent ascertained and documented? It is a sensitive point. Questions are often asked In summing up this review My reviewers assessment is that:  1. My view is to regard the manuscript under review to BMJ Open to be a document deserving a general overview review of its qualities in presenting study population, the procedures in data collection and editing of tables 2 A statistical review is also well placed The study raises as a prominent conclusion the aspect of community engagement as a potentially promotive factor in environmental health promotion and support of mental wellbeing also with regard to area deprivation.
--	---

REVIEWER	Saavedra, Javier University of Seville
REVIEW RETURNED	09-Feb-2021

GENERAL COMMENTS	This manuscript is a valuable contribution in relation to the association between “community cultural engagement” and health variables. Among other strong points, I stress on the size of the sample and the analysis of the moderation of the deprivation variable of the areas. However, there are important aspects that must be discussed, and perhaps reformulated before possible publication. Running an OLS regression requires checking a series of assumptions, the best known are: variance of the errors should be consistent for all observations; lack of multicollonearity; error observations must not predict next observations, etc. The authors
---

	do not make any comment regarding the verification of these or other premises in the execution of the regression. The instruments used to evaluate mental health are really screenings instruments. I think this is a weakness. I think it is necessary to point out this fact as a weakness. Especially the life satisfaction scale since it seems to be an ad-hoc scale that has not been validated. am I right? No minimum data on the psychometric validity of these instruments (internal consistency, reliability ...) are provided. It would be interesting to know the internal consistency of the total scale of these instruments in the investigation. The results show a significant association between CCO and health. However, nothing is said about the intensity of this association (effect size). Considering such a large sample, it is not surprising that we have significant associations. It seems that effect sizes, taking into account the coefficients, are not high. I would like the authors to explicitly discuss this fact. In this sense, it would be interesting to know the total explained variance of the models. It is not clear to me if the data used in each of the waves is the same. In the first analyzes, CCO data are taken from the wave 2 and the health data from wave 5. But in the sensitivity analysis, the use of the data is the opposite. In any case, I doubt that the study can be called longitudinal and, as the authors say in the limitations, it is not possible to perform any causal analysis. This warning should be made much earlier in the article, perhaps in the abstract, not the first time in limitations at the end of the paper. Why do authors think that they do not find any moderation of deprivation using linear continuous measure of deprivation (decile rank) and they do with categorical measures? The sensitivity analysis is not explained in method. In order to hypothesize some causal relationship between CCO and health, I would expect an increase of the intensity of the slope of the equation that predicted health in the wave 5 compare to the slope of the equation that predicted health in the contrary direction. This maybe could indicate some causal effect. Otherwise, results only show that there is association between health and CCO adjusted sociodemographic variables and that it seems that there is an interaction with deprivation areas, what is not little. I would like to discuss these questions with authors.
--	---

REVIEWER	Tabassum, Faiza University of Southampton, School of Social Sciences
REVIEW RETURNED	06-Mar-2021

GENERAL COMMENTS	This paper examines the associations between community cultural engagements (CCE) and well-being and how these associations moderated by the area deprivation. This paper uses a nationally representative longitudinal data. There are number of issues which need to be addressed by the authors. Being a reader, I found this paper not an easy read particularly in terms of the length of analyses presented here. This paper has used three different outcome variables; two exposure variables and one moderator (IMD) in a continuous as well as a categorical variable. The authors may consider of computing some kind of index by combining cultural events and museums visits together so the results could be more meaningfully presented. Longitudinal analyses: the authors have repeatedly mentioned that their results are based on the longitudinal analyses. However, I am
--

	unable to see any longitudinal analyses actually done. The authors have mentioned OLS regression which they have used and OLS is not a longitudinal analysis technique. If any longitudinal analysis is done then it needs to be specified clearly. This is actually my biggest concern on this paper is that the researchers have used a longitudinal data but unable to conduct a proper longitudinal analyses. It is mentioned in the Statistical analysis section that sequentially constructed OLS regression models are used. However, no further description is provided eg how are these models are constructed and why an interaction term is used? What is the logic of formulating models in this way? How much the variation has been explained by each block of models? Tables 2 and onward: it is not clear what are these models eg 'Basic Model (what does it include); then + IMD; Demographic factors; socioeconomic position. It should be made it clear in the statistical analysis section and in the tables' footnotes. My question is that after running the basic model, IMD term is added in the regression equation? Then what is the IMD term under CCE in the first column? The whole model construction is very confusing to understand, it needs to be clearly stated in Statistical Analysis section. Table 1 does not seem to make any contribution in this research. Instead, my suggestion is to have a table which reports the associations between eg each of the well-being measures and CCE, IMD and all the explanatory variables. Some more description of data is required for example, which years these waves were associated and why particularly these waves were used in this paper? Figures: it is not clear that which results are used to plot these graphs? Variables: I found it hard to figure out which variables are from wave 2 and which are from wave 5. Better description of the GHQ 12 needed so reader knows what a high or low score actually means. Further evidence needed for why GHQ 12 is a good stand in for mental well-being, which is quite a multi-faceted concept. Generally, all three outcome variables need more description and references. CCE: some more detail how these variables are constructed and what is their range? IMD: which particular year? Results: generally in the present form, the tables are very crowded and as a result hard to understand. The authors should think carefully of condensing the analyses and presenting them in a way easy to understand.
--	---

	There is no explanation of the regression coefficients, eg it is not enough to say that the coefficient is positive or negative, the researchers need to mention how large or small the coefficients in comparison to previous studies. For example, the coefficient of life satisfaction for the interaction (CCE * 20% most deprived) is 0.04; how meaningful is this number associated with life satisfaction? There are some results which need the attention of the authors particularly the interaction of CCE and IMD. Eg, in case of SF-12 when IMD is used as a continuous variable the coefficient is -0.01 but it is +0.03 when IMD is a categorical var (20% most deprived). My worry is that the sign of regression coefficient has changed, what is the explanation of this? Most importantly, all such associations (interactions between CCE*IMD) were shown only in case when CCE representing the cultural activities but not the museums. Therefore, this point needs to be highlighted. At the current form, most of the explanation does not seem to reflect the statistical results particularly the 'numbers' rather it feels like a 'speculation' at a number of occasions in the Discussion and Conclusion sections. It needs to be spelled out that no associations between GHQ and CCE*IMD were found, what does it mean? But at the same time SF-12 shows some associations. What are the policy implications of these results?
--	---

VERSION 1 – AUTHOR RESPONSE

Reviewer 1 - Dr. Ann Hemingway, Bournemouth University Comments to the Author:

Thank you for this very interesting and important paper, I just have some minor concerns regarding the title, and the measures analysed. You mention wellbeing in your title and then mention mental wellbeing in your abstract. The measures you have focused your analysis on are life satisfaction, mental distress and mental health functioning. It may be more appropriate to either clarify your focus in your title or build some links in the paper between these measures and overall definitions of wellbeing. Overall a really interesting and relevant paper thank you.

Thank you, we are pleased that you found this very interesting and important paper. We have now clarified this in our title.

Reviewer 2 - Dr. Peter Westerholm, Uppsala University, Home address Comments to the Author:

The authors have picked on an interesting subject to inform colleagues of Medical Professions on the links between involving health professionals on community cultural assets and programmes and subsequent wellbeing and lifelong life satisfaction of subjects / citizens taking visiting museums and heritage sites into account and also qualitative conditions of neighbourhood deprivation as observed by the authors. The document submitted to The BMJ Open aims to describe a study addressing the possible association between engagement of a study population in community cultural assets such as attendance at cultural events, visiting museums and heritage sites and subsequent well-being (life satisfaction, mental distress and mental health functioning exploring the question whether such effects may be moderated by area neighbourhood deprivation. The study population of this undertaking is a UK general adult population with a total sample of 17, 948 subjects. It is based on a 3- year study based on Using an OLS regression analytic technique the authors observed that positive engagement of study population resulted in improvements of mental wellbeing and subsequent life satisfaction. At the same time a negative association was observed of such cultural association and health endpoint Mental distress in the study population. Basically three well-being indicators are to be explored – Life satisfaction, mental distress and mental health functioning. The authoring group has at the same time observed supportive evidence for study populations recruited from

deprived areas – implying a combination of multiple Deprivation components with weakness of combined components of Living Environment , Income and Employment Deprivation, used as a proxy of neighbourhood impact. So, in authors own assessment this study shows that engagements in cultural community assets are associated with improvement of well-being of population studied. The subject of deprivation of populations or areas involved in framing of research question(s) is certainly an innovative aspect in projects of this type. There is only scarce documentation available on this subject. The manuscript is based on data drawn from the database of the UK Household Longitudinal Study (UKHLS) community.

Reviewers´ overall assessment of manuscript scientific relevance and qualities is favourable in observing its demonstration of engagement in cultural aspects in a study population to be positively associated with improvement of life satisfaction and perceptions of mental health functioning and, conversely, negatively associated with signs indicating mental distress. To reviewer the manuscript as presently drafted opens up an aspect of editing contents to be attractive to readers of BMJ Open access Research Journal.

Thank you, we are very grateful for your very positive comments.

The ideal solution to me to edit draft texts to offer clearly distinctive components of “Introduction”, “Background” moving over to “Research questions/ Issues” and from there on to “Study Population(s)”, “Information sources” and an overview of “Methods” describing the mechanics of how it was all managed including a time schedule of management of study as actually prepared and performed deserves a concentrated effort The section “Statistical Methods” is an obvious prelude to presenting “Results”. It is certainly not my preference to be identified as orthodox in editing of scientific draft manuscripts for publishing. I did, however, observe that my first impression when reading submitted manuscript was to react in seeing need of an experienced hard-nosed editors hand to go through text in making it effective and attractive to readers, in avoiding repetitions and redundancies in explanations.

My spontaneous comments/questions follow here:

1. **Abstract.** The abstract is to impeccable in its concentration on Life satisfaction, mental health function possible.ning and mental stress and giving some selected results.
2. **Commendable emphasis of study being based on nationally representative population with 3-year FU period**
3. **Necessary to claim causality as not being possible to establish? Reviewer: Yes – but it may still be suggestive**

We are unsure if these remarks were intended for us or for the reviewer. It is unclear what changes are being requested. However, we have addressed the points below where changes are clearer.

4. **Important Confounding factors controlled Is this Correct ? Which CF:s have been examined and assessed by authors?**

The set of confounding factors controlled in the models has been stated in the Measures section, which included regional locations, demographic backgrounds and socio-economic factors.

5. **Ethics: Was improved consent by study population ascertained at any stage? How was this done? Which were the questions asked?**

The ethics of the data from Understanding Society: The UK Household Longitudinal Study (UKHLS) were approved by the University of Essex Ethics Committee, where respondents aged 16 or above provided written consent to participate. UKHLS is one of the largest nationally-representative study in the UK, which its data have been widely used by researchers and the government. For more information on the study and its ethics procedure, see <https://www.understandingsociety.ac.uk/documentation/mainstage/user-guides/main-survey-user-guide/ethics>

6. **Tables 1 – 2 – 3 – 4**

All these tables require explanatory texts for benefit of readers. Where is this information to be included in manuscript for conclusive review? For proper understanding of authors intention regarding association between community cultural engagement (CCE) and mental health distress (GHQ-12) at wave 5 of authors intentions and implications in reference made to “OLS regression estimating the association “ For tables 2-3-4 there is a need to explain “notes on IMD (ref: medium) 20% deprived 20% least deprived.....etc.

Thank you for your suggestion. As suggested by Reviewer 4, we have now rearranged the tables. In Tables 2-4, we now only present results with the continuous measure of IMD. Results with alternative measures for IMD, the 10% and 20% thresholds (i.e. 10%/20% most deprived, medium levels of area deprivation, 10%/20% least deprived), are now presented in the Supplementary Materials. All tables now include information on the models:

“Note: M1 (basic model) included CCE. M2 additionally controlled for Index of Multiple Deprivation (IMD) and interaction terms (CCE*IMD). M3 additionally adjusted for demographic factors (age, gender, ethnicity, marital status, presence of children under age 16, whether or not living alone, regional locations). M4 additionally adjusted for socio-economic position (educational levels, occupational socio-economic status, household monthly gross income, and housing tenure).”

To reduce confusion, we have pared down the supplementaries showing results estimating the association between CCE and the outcome variables measured at Wave 2 and mentioned that materials are available upon request.

For the benefit of readers of eventually published document.

For figures 1 and 2 there is in the submitted draft manuscript package a draft said to be a figure legend explaining figures. I have not found it in manuscript under review

We are very sorry to hear that you were unable to find Figures 1 and 2. These figures should be now included with the submission.

- 7. I see that the references to bibliography are assembled as a plain listing of authors names , documents referred to and bibliographical details. Is this in pursuance of BMJ Open practices ?**

We believe the reference style is in line with the BMJ Open practices.

Conclusive reviewers assessment:

- 8. This is an interesting manuscript. It deserves to be taken seriously. Its main and principal value lies in an aspect of originality in observing findings supportive of suggesting area of study population (residential area or work area ?) to be capable of “moderating” effects of communal cultural engagement CEE, for instance with regard to end-point variables mental health functioning and prevalence of mental distress. This observation is at the present time of suggestive nature.**

Thank you for your positive comments. The focus of area is respondents' residential areas, rather than work area. This has already been stated in the last paragraph of the Introduction:

“In particular, given that differential exposure to risks affects health differently, we explored whether individuals living in areas of high deprivation who are at higher risks of experiencing poorer mental health gain the same or even greater wellbeing benefits from CCE as individuals living in areas of low deprivation.”

- 9. A challenging observation of authors is also the somewhat paradoxical observation in context of the study that subjects residing in areas of high deprivation may benefit more from such engagement, opening up a discussion of validity of place-based investment schemes in areas of high deprivation. This observation needs to be qualified and verified in other studies to be regarded as established, reproducible and valid in assessing strategies of place-based schemes aiming at health promotion. The manuscript needs a hard-nosed Editors editing examination to go through text as presently drafted including texts of tables and figures. Reviewers recommendation is to include a separate statistical review of analyses and methods.**

We have now stated in Conclusion that findings from this study warrant further research in other datasets.

- 10. The main study is based on following up of a study population. It is therefore necessary to tell readers about how this was done and its completeness. Were there losses to the follow-up? Have the losses to Follow-ups of the study population affected conclusions or interpretation of authors findings?**

Thank you. We have now included a brief description of how our sample was extracted from Waves 2 and 5 from UKHLS data in the Data & Methods and Results sections:

“In order to investigate the role of area deprivation, we used geo-coded UKHLS data in which participating households’ addresses have been positioned into a number of spatial zoning systems (e.g. administrative and census statistical geographies). For our analysis, we extracted a sample of adults living in England who responded to the Wave 2 (2010/12; response rate=84%), where data on engagement in CCE were first available, and Wave 5 (2013/15; response rate=85%) interviews. We only considered respondents who completed both Waves 2 and 5 interviews and those who answered across all measures, as well as respondents with a valid sampling weight value”. (Data & Methods)

“In our sample, 38,069 participants living in England responded to the Wave 2 and 30,635 participants responded to the Wave 5 interviews. 25,464 individuals completed both interviews. Of these, 23,247 participants provided answers for the outcome variables, with 23,244 respondents additionally answered questions on CCE, and 22,463 individuals answered across all other measures. Amongst them, 14,833 received a non-zero Wave 2 longitudinal weight value provided by UKHLS. Of these, all but 50 (14,783) received a valid tailored weight value for this analysis.”. (Results).

When considering respondents with a valid weight, there were around 14,833 respondents with available data and, of those, 14,783 people have answered across all measures. Given that the N missing is trivial, we have decided to remove the multiple imputations (which were used to impute missingness). The results remain materially unchanged.

I am here adding a general comment on the submitted manuscript and the tables presenting the material and the analyses.

11. General comment on tables 1 through 4

The contents of all four tables consist of presentations of findings and observations characterizing study populations. Imputation of material to be complementary has apparently been carried out. Details on this procedure explaining what and how this was actually done not clarified in manuscript as presently drafted.

Thank you for your comment. We have now removed the multiple imputations due to low levels of missingness (please see the response above).

12. In table 2 OLS regression analyses are presented on CCE and life satisfaction weighted at wave 5. Observations given of regression as interaction terms, socioeconomic positions and demographic terms. Under rubric “statistical analyses” reference is made to confounding factors and their construction based on their use in predictive models of analysis. Which confounding variables were taken into account? How was this control actually implemented?

Details on confounding variables have been stated in the paragraph above, which included broad regional variations (North, Midlands, South) as well as demographic (e.g. age, gender and ethnicity) and socio-economic (e.g. educational levels, household income and occupational socio-economic status) variables measured at Wave 2.

In our analysis, we built OLS regression models by adding exposures and confounding variables sequentially. In Model 1 (basic model), we included only the CCE. Model 2 additionally added Index of Deprivation (IMD) and the interaction terms (i.e. CCE*IMD). Model 3 additionally included demographic factors, and finally Model 4 additionally adjusted for socio-economic position. We have now included the information in the manuscript, as well as the tables.

13. Under rubric “Strengths and limitations of this study” is stated that in this observational study “causality can not be established”. In reviewer’s view this general statement on possibility to establish causality in a statistical association seen in an observational study seems me to be redundant. Is it really needed? The issue in itself may indeed deserve a comment. We know the existing differences of opinion on this issue in our own professional circles. On grounds of evidence in this manuscript and also in other studies evidence of suggestive nature may well be construed.

Your comment on the causality statement is very well-received. However, we think it is necessary to highlight that, whilst our analysis is based on longitudinal data, causality cannot be established given that it is an observational study where some other unobserved factors may still confound the relationship between CCE and mental health/wellbeing. We have now explained this in the section:

“First, causality cannot be established given that this is an observational study where other unobserved factors may still confound the association between CCE and mental health/wellbeing. The relatively low levels of model fit and fairly small effect sizes make it worthwhile for further research to test how additional socio-demographic measures not routinely collected by social surveys might shape well-being outcomes.”

14. **As to acceptability of manuscript as presently drafted for BMJ Open publishing I recommend to have it examined in a statistical review of statistical methods and techniques used, in techniques used in supplementing study population with imputed subjects and groups and also in drafting of explanatory notes and comments to tables presenting results.**
15. **Reference made to factors confounding observed associations of engagement in community cultural assets deserve special attention.**
We agree and we discuss the confounders we selected for the model in the methods section.
16. **This manuscript now reviewed has been approved in UK HOUSHOLD (UKHLS) in addressing main study, innovation panel waves and informed consent for all data linkages excepting health records. This is an excellent move.**
Thank you for your very positive comments.

Reviewers question:

17. **How was this informed consent ascertained and documented? It is a sensitive point. Questions are often asked In summing up this review.**
Thank you. As answered above, the ethics of the data from Understanding Society: The UK Household Longitudinal Study (UKHLS) were approved by the University of Essex Ethics Committee, where respondents aged 16 or above provided written consent to participate. Please see our response above.

My reviewers assessment is that:

18. **My view is to regard the manuscript under review to BMJ Open to be a document deserving a general overview review of its qualities in presenting study population, the procedures in data collection and editing of tables. A statistical review is also well placed. The study raises as a prominent conclusion the aspect of community engagement as a potentially promotive factor in environmental health promotion and support of mental wellbeing also with regard to area deprivation.**
Thank you for the very positive comments. We are very grateful for your time to review the paper.

Reviewer 3 - Dr. Javier Saavedra, University of Seville Comments to the Author:

This manuscript is a valuable contribution in relation to the association between “community cultural engagement” and health variables. Among other strong points, I stress on the size of the sample and the analysis of the moderation of the deprivation variable of the areas. However, there are important aspects that must be discussed, and perhaps reformulated before possible publication.

1. **Running an OLS regression requires checking a series of assumptions, the best known are: variance of the errors should be consistent for all observations; lack of multicollinearity; error observations must not predict next observations, etc. The authors do not make any comment regarding the verification of these or other premises in the execution of the regression.**
Thank you for raising this concern. We have now provided a more detailed description on OLS assumption checks and have been more cautious in mentioning possible problems with the less satisfactory diagnostic checks for models estimating life satisfaction. We are happy to provide diagnostics to reviewers or include them in supplementary material if this would be useful. We

have not done that yet because other reviewers' comments suggested a need to reduce the volume of material presented:

"To check whether our data met the assumption of OLS regressions, we ran a series of regression diagnostic tests. Our tests show that the distribution of residuals was almost homoscedastic and normal for models estimating mental distress and mental health functioning. The distribution was less satisfactory for models estimating life satisfaction, most likely due to the more discrete scale of this variable. However, our large sample size and the fact we present weighted estimates with robust standard errors should mitigate against problems arising from model fit. Nonetheless, more caution should be exercised when interpreting the results of the life satisfaction models as compared with the other well-being outcomes. The risk of multicollinearity was also very low with a mean Variance Inflation Factor (VIF) of 2.03 among the independent variables."

- 2. The instruments used to evaluate mental health are really screenings instruments. I think this is a weakness. I think it is necessary to point out this fact as a weakness. Especially the life satisfaction scale since it seems to be an ad-hoc scale that has not been validated. am I right? No minimum data on the psychometric validity of these instruments (internal consistency, reliability ...) are provided. It would be interesting to know the internal consistency of the total scale of these instruments in the investigation.**

Thank you for your comments. They are very well-received. However, we do not agree that GHQ-12, SF12 and life satisfaction were a serious weakness. This study is based on nationally-representative data, and these three mental health/wellbeing measures are commonly used to measure wellbeing amongst a general population (GHQ: $\alpha=0.91$; SF-12: $\alpha=0.90$). For life satisfaction, whilst the scale may appear ambiguous to some extent, it has been used as one of the four questions to measure personal wellbeing in the UK general population by the UK Office for National Statistics (ONS) despite the scale varies slightly (the scale from ONS ranges from 1-10 whereas the scale from the UKHLS data ranges from 1-7). To justify our choice of using these measures, we have now cited references in the manuscript, as well as suggesting in the strengths and weaknesses section that future research is needed to explore other mental health measures that are more commonly used in clinical practices:

"Future research may also want to explore other mental health measures that are more commonly used in clinical practices, such as the Patient Health Questionnaire (PHQ) and General Anxiety Disorder (GAD) measure."

We have now also included Cronbach's alpha for both GHQ-12 and SF-12 in the manuscript to test the internal consistency:

"The GHQ-12 self-reported questionnaire includes 12 4-point items (such as sleeping problems, overall happiness and depressive symptoms; $\alpha=0.91$). The scale was computed additively, ranging from 1 to 4; higher scores indicating a greater incidence of mental distress. Mental health functioning was measured using SF-12 (12-Item Short Form Health Survey; $\alpha=0.90$); a well-validated survey that was designed to measure respondents' general health-related quality of life."

- 3. The results show a significant association between CCO and health. However, nothing is said about the intensity of this association (effect size). Considering such a large sample, it is not surprising that we have significant associations. It seems that effect sizes, taking into account the coefficients, are not high. I would like the authors to explicitly discuss this fact. In this sense, it would be interesting to know the total explained variance of the models.**

Thank you for your valuable comment. We totally agree with you. We have now included beta in the manuscript (beta was not presented in the tables to avoid crowdedness) and R-squared in the tables to shed lights on the effect sizes and the total explained variance of the models. We have also discussed this in the Discussion section:

"Importantly, this paper found that such associations were independent of individuals' demographic background, socio-economic characteristics and regional locations. In particular, our models show that every one standard deviation increase in CCE is associated with higher life satisfaction and mental health functioning (by 0.06-0.13 standard deviations) and lower mental distress (by 0.05-0.09 standard deviations). Although the magnitude of these effects is fairly small, such associations were evident even after considering levels of area deprivation, demographics, and socio-economic

factors and when predicting the outcomes measured after three years. This suggests that, while social and geographical factors can influence engagement rates [12–14,18,39], CCE is consistently associated with minor improvements in mental health and wellbeing regardless of where people live.”

We have also flagged up the relatively low levels of model fit as a limitation and it is perhaps worthwhile for further research to test how additional socio-demographic measures not routinely collected by social surveys might shape well-being outcomes.

- 4. It is not clear to me if the data used in each of the waves is the same. In the first analyzes, CCO data are taken from the wave 2 and the health data from wave 5. But in the sensitivity analysis, the use of the data is the opposite. In any case, I doubt that the study can be called longitudinal and, as the authors say in the limitations, it is not possible to perform any causal analysis. This warning should be made much earlier in the article, perhaps in the abstract, not the first time in limitations at the end of the paper.**

Thank you for your comments. We understand that there has been some confusion due to the volume of material. We have therefore removed the Wave 2 analysis and only mentioned it briefly. Results are available on request.

We have also revised the abstract and stated that causality claim cannot be generated from our results: “Given that causal mechanisms were not tested, causal claims cannot be generated from the results”.

- 5. Why do authors think that they do not find any moderation of deprivation using linear continuous measure of deprivation (decile rank) and they do with categorical measures?**

To respond to other reviewers’ comments we have re-done the analysis by using the original data, instead of imputed data, due to a trivial sample reduction after considering respondents with a valid weight value. We have also re-scaled the GHQ-12 and SF-12 scores by using the original, unstandardised scales, and found some moderation in the association between both types of CCE and GHQ-12 and SF-12 when using the linear continuous measure of deprivation. The moderation associations were largely replicated when using the categorical measures, although it was less prominent with the 20% threshold for SF-12 outcome. Also, no moderation was found when using the three-fold measures (both 10% and 20%) for mental distress.

- 6. The sensitivity analysis is not explained in method. In order to hypothesize some causal relationship between CCO and health, I would expect an increase of the intensity of the slope of the equation that predicted health in the wave 5 compare to the slope of the equation that predicted health in the contrary direction. This maybe could indicate some causal effect. Otherwise, results only show that there is association between health and CCO adjusted sociodemographic variables and that it seems that there is an interaction with deprivation areas, what is not little.**

We have simplified the manuscript by removing the complex discussion of sensitivity checks. So this problem should have been removed.

- 7. I would like to discuss theses questions with authors.**

Thank you for your valuable comments and suggestions. We hope that all issues have now been addressed.

Reviewer 4 - Dr. Faiza Tabassum, University of Southampton Comments to the Author:

- 1. This paper examines the associations between community cultural engagements (CCE) and well-being and how these associations moderated by the area deprivation. This paper uses a nationally representative longitudinal data. There are number of issues which need to be addressed by the authors. Being a reader, I found this paper not an easy read particularly in terms of the length of analyses presented here. This paper has used three different outcome variables; two exposure variables and one moderator (IMD) in a continuous as well as a categorical variable. The authors may consider of computing some kind of index by combining cultural events and museums visits together so the results could be more meaningfully presented.**

Thank you for your comment. While we agree that combining cultural events and museums and heritage visits might help make the results presented meaningfully, we feel these two activities are

quite different. For instance, many cultural events require monetary resources e.g. fees of entry for musical, theatres and galleries and can often be found in local communities, whereas museums and heritage sites tend to be more distal from residential areas and most of the museums are free of charge. In addition, our analysis shows different moderation associations between these two types of CCE, suggesting that collapsing them into one variable might not be appropriate.

- 2. Longitudinal analyses: the authors have repeatedly mentioned that their results are based on the longitudinal analyses. However, I am unable to see any longitudinal analyses actually done. The authors have mentioned OLS regression which they have used and OLS is not a longitudinal analysis technique. If any longitudinal analysis is done then it needs to be specified clearly. This is actually my biggest concern on this paper is that the researchers have used a longitudinal data but unable to conduct a proper longitudinal analyses.**

We are very sorry for the confusion and have now removed the term “longitudinal association” from the manuscript.

- 3. It is mentioned in the Statistical analysis section that sequentially constructed OLS regression models are used. However, no further description is provided eg how are these models are constructed and why an interaction term is used? What is the logic of formulating models in this way? How much the variation has been explained by each block of models?**

The Statistical analysis section has now included an explanation for model construction and the rationale behind the construction:

“To understand whether the relationship between CCE (x) and mental wellbeing measures (y) varied with area deprivation (the potential moderator), we ran a cross-sectional analysis using OLS regression models. Given that residential location is highly correlated with personal demographic and socio-economic factors, the regression models were constructed sequentially to understand the changes of the association between CCE and mental wellbeing. In Model 1 (basic model), we included only CCE. Model 2 additionally controlled for IMD and the interaction terms (i.e. CCE*IMD). Model 3 additionally included demographic factors, and finally Model 4 adjusted for socio-economic position”.

The rationale of adding an interaction term has already been described in the Introduction section. For instance:

“Understanding whether there is any moderation is crucial and relevant to current public health strategies and interventions such as ‘social prescribing’ schemes and place-based funding streams for the cultural sector. These are predicated on the belief that increasing the local availability of assets and their usage could lead to increased CCE and thus improved health outcomes [27–32]. But it remains unclear whether investment in cultural assets in different locations holds equal potential for positively influencing health.”.

R-squared has also been included in the tables to show the proportion of the variance explained by the model for each block.

- 4. Tables 2 and onward: it is not clear what are these models eg ‘Basic Model (what does it include); then + IMD; Demographic factors; socioeconomic position. It should be made it clear in the statistical analysis section and in the tables’ footnotes.**

Thank you. We have now provided a description of the models in the Statistical analysis section and in the tables’ footnote (please see response to Q3).

My question is that after running the basic model, IMD term is added in the regression equation? Then what is the IMD term under CCE in the first column? The whole model construction is very confusing to understand, it needs to be clearly stated in Statistical Analysis section.

Thank you. We have now provided a description of the models in the Statistical analysis section and in the tables’ footnote (please see response to Q3).

The IMD term under CCE show the coefficients of IMD when CCE=0.

5. **Table 1 does not seem to make any contribution in this research. Instead, my suggestion is to have a table which reports the associations between eg each of the well-being measures and CCE, IMD and all the explanatory variables.**

Thank you for your suggestion. We have now re-done the analysis by using the available data, instead of the imputed data due to a small sample reduction rate after considering respondents with a valid weight value. Therefore, we have now provided descriptive statistics of both weighted and unweighted samples.

6. **Some more description of data is required for example, which years these waves were associated and why particularly these waves were used in this paper?**

The years the waves were associated have already been described in the Data and Methods section (i.e. CCE and covariates were measured at Wave 2 (2010/12) and outcome variables were measured at Wave 5 (2013/15). Wave 2 was used as data on engagement in CCE were first available, and we tracked for 3 years to balance out the need for a longitudinal outcome measure with the problem that attrition and unknown changes in CCE increase over longer timeframes. Hence, we arrived at a 3 year follow up as the best compromise.

7. **Figures: it is not clear that which results are used to plot these graphs?**

Results with significant interaction terms were also presented in graphs for interpretation. We have added Figure notes next to those results. For example:

“When comparing with the most distinct cultural attendance frequency (i.e. at least once a week vs none in the past 12 months), the score of mental distress amongst people living in the 10% least deprived areas was decreased from 1.89 (no engagement) to 1.83 (weekly engagement). Similarly, of people living in the 10% most deprived areas, their mental distress score was decreased from 2.05 (no engagement) to 1.89 (weekly engagement). The differences in mental distress scores between people living in varying levels of area deprivation became smaller with increased cultural attendance frequency (Figure 1)”.

Variables:

8. **I found it hard to figure out which variables are from wave 2 and which are from wave 5.**

Thank you for your comment. Where the variables were derived from has already been stated in the manuscript. We have also explained that the 2011 LSOA data were matched with Wave 2 UKHLS where the data were collected in 2010/12:

“We defined neighbourhoods as 2011 census Lower Layer Super Output Areas (LSOA) and matched the 2011 LSOA with Wave 2 UKHLS where data were collected between 2010/12”;

“CCE was measured in Wave 2”;

“We explored three outcome wellbeing measures in Wave 5, which took place around three years after our Wave 2 baseline. These measures were life satisfaction, mental distress and mental health functioning.”; and

“In our analysis, we controlled for broad regional variations (North, Midlands, South) as well as demographic and socio-economic variables in Wave 2”.

9. **Better description of the GHQ 12 needed so reader knows what a high or low score actually means. Further evidence needed for why GHQ 12 is a good stand in for mental well-being, which is quite a multi-faceted concept. Generally, all three outcome variables need more description and references.**

Thank you for your comment. We have now included more description and references on the outcome measures. We, however, are reluctant to add a cut-off point for GHQ 12 given that the variable was not used as categorical in our analysis. Instead, we have now re-scaled the variable by un-standardising it and reported the range, mean and SE in Table 1. We believe this should provide readers an understanding of the average GHQ-12 score within the sample.

10. **CCE: some more detail how these variables are constructed and what is their range?**

We have now added more detail on the construction of CCE variables. The range has already been stated in the manuscript and Table 1:

“For CCE, we focused on attendance at cultural events; including going to the theatre, concerts, opera and exhibition, and museums/galleries and heritage sites visits (a full list of CCE is provided

in Appendix I). CCE was measured in Wave 2. At this wave, respondents were asked how often they had attended any of the cultural events, visited museums/galleries and visited heritage sites in the past 12 months. Frequency of engagement with these activities was categorised as “not once in the last 12 months”, “once in the last 12 months”, “twice in the last 12 months”, “less often than once a month but at least 3 or 4 times a year”, “less often than once a week but at least once a month” and “at least once a week”. Due to the similar nature of the activities, visits in museums/galleries and heritage sites were collapsed into one variable. Both types of CCE (cultural attendance and museum and heritage engagement) were treated as continuous measures.”.

11. IMD: which particular year?

IMD was collected in 2015, which has already been stated in the Measures section.

12. Results: generally in the present form, the tables are very crowded and as a result hard to understand. The authors should think carefully of condensing the analyses and presenting them in a way easy to understand.

Thank you for your comment. We are aware the tables were crowded and have therefore decided to only present the main results with the continuous measure of IMD. Results using the alternative measures (i.e. the three-fold variables using the 10%/20% threshold) are now presented in Supplementaries.

13. There is no explanation of the regression coefficients, eg it is not enough to say that the coefficient is positive or negative, the researchers need to mention how large or small the coefficients in comparison to previous studies. For example, the coefficient of life satisfaction for the interaction (CCE * 20% most deprived) is 0.04; how meaningful is this number associated with life satisfaction?

It is likely that different models with various sets of community cultural engagement and covariates measures were used in previous studies, therefore it becomes challenging to compare the magnitude of coefficients. However, we have now included beta in the manuscript and R-squared to shed lights on the effect sizes and the total explained variance of the models. We have also discussed this in the Discussion section:

“Importantly, this paper found that such associations were independent of individuals’ demographic background, socio-economic characteristics and regional locations. In particular, our models show that every one standard deviation increase in CCE is associated with higher life satisfaction and mental health functioning (by 0.06-0.13 standard deviations) and lower mental distress (by 0.05-0.09 standard deviations). Although the magnitude of these effects is fairly small, such associations were evident even after considering levels of area deprivation, demographics, and socio-economic factors and when predicting the outcomes measured after three years. This suggests that, while social and geographical factors can influence engagement rates [12–14,18,39], CCE is consistently associated with minor improvements in mental health and wellbeing regardless of where people live.”.

To help interpret the coefficients of the interaction terms, we have now estimated the marginal effects and provided an alternative explanation for the results. For instance, for mental health functioning:

“Of those living in the 10% least deprived areas, their score in mental health functioning was increased from 3.79-3.80 (no engagement) to 4.07 (weekly engagement) for both types of CCE. Amongst those living in the 10% most deprived areas, their score was increased from 3.39-3.44 (no engagement) to 3.85-3.86 (weekly engagement). The differences in mental health functioning scores between people living in varying levels of area deprivation became narrower with increased CCE (Figures 2 & 3)”.

Results with significant interaction terms have also been presented in graphs for data visualisation.

14. There are some results which need the attention of the authors particularly the interaction of CCE and IMD. Eg, in case of SF-12 when IMD is used as a continuous variable the coefficient is -0.01 but it is +0.03 when IMD is a categorical var (20% most deprived). My worry is that the sign of regression coefficient has changed, what is the explanation of this?

We have now re-run the analysis. The issue of different coefficient signs no longer appears. Nonetheless, the different signs for interactions could be due to various measures of IMD. The effects of interactions are likely to be different depending on the group which the main effect is interacting with. Further, the interpretation of the interaction terms also changes between continuous*continuous and continuous*categorical. To help interpret the interaction terms, graphs and predicted values obtained from marginal effects are often suggested.

15. **Most importantly, all such associations (interactions between CCE*IMD) were shown only in case when CCE representing the cultural activities but not the museums. Therefore, this point needs to be highlighted. At the current form, most of the explanation does not seem to reflect the statistical results particularly the ‘numbers’ rather it feels like a ‘speculation’ at a number of occasions in the Discussion and Conclusion sections.**

In the new analysis, moderation between CCE and IMD is shown in both cultural events attendance and museum and heritage engagement for SF12 outcome, although moderation for GHQ-12 outcome was only shown for cultural attendance. We have now explained the associations in the Discussion section:

“No moderations are found for life satisfaction, nor for engagement in museum and heritage and mental distress. This suggests that the benefits of CCE on life satisfaction and mental distress are similar regardless of residential locations.”.

We have now provided an explanation to reflect the statistical results in the Discussion section:

“In particular, our models show that every one standard deviation increase in CCE is associated with higher life satisfaction and mental health functioning (by 0.06-0.13 standard deviations) and lower mental distress (by 0.05-0.09 standard deviations). Although the magnitude of these effects is fairly small, such associations were evident even after considering levels of area deprivation, demographics, and socio-economic factors and when predicting the outcomes measured after three years. This suggests that, while social and geographical factors can influence engagement rates [12–14,18,39], CCE is consistently associated with minor improvements in mental health and wellbeing regardless of where people live.”.

16. **It needs to be spelled out that no associations between GHQ and CCE*IMD were found, what does it mean? But at the same time SF-12 shows some associations. What are the policy implications of these results?**

In our new analysis, there were moderations between CCE*IMD and GHQ for cultural event attendance, but not for museum and heritage engagement. We have also found moderations for both types of CCE*IMD and SF-12. We have now discussed this in the Discussion section:

“In particular, we found that the rates of growth in mental health functioning that accompany CCE are stronger amongst people living in deprived areas, and that the rates of decline in mental distress that accompany cultural events attendance are also more prominent amongst those living in deprived areas”.

“No moderations are found for life satisfaction, nor for engagement in museum and heritage and mental distress. This suggests that the benefits of CCE on life satisfaction and mental distress are similar regardless of residential locations.”.

Whilst we have found some moderations in the analysis, the moderations are less consistent across all outcome measures. To avoid over-interpreting the results, we feel the current form of Conclusion section suffices:

“Further, there is also some evidence of moderation, with individuals in areas of high deprivation potentially even able to benefit more from CCE in terms of mental health functioning and improvements in mental distress. However, this does not mitigate the problem that individuals in areas of high deprivation are less likely to engage in CCE. This, therefore, suggests the importance of exploring further the effects of place-based funding schemes that involve investment in areas of higher deprivation to improve engagement rates to confirm if such schemes could help to promote higher levels of wellbeing amongst individuals in such areas.”.

STROBE Statement—Checklist of items that should be included in reports of cross-sectional studies		
	Recommendation	Page/Line number
Title and abstract		
	(a) Indicate the study's design with a commonly used term in the title or the abstract	P1
	(b) Provide in the abstract an informative and balanced summary of what was done and what was found	P2
Introduction		
Background/rationale	Explain the scientific background and rationale for the investigation being reported	P4
Objectives	State specific objectives, including any prespecified hypotheses	P5
Methods		
Study design	Present key elements of study design early in the paper	P5
Setting	Describe the setting, locations, and relevant dates, including periods of recruitment, exposure, follow-up, and data collection	P5
Participants	(a) Give the eligibility criteria, and the sources and methods of selection of participants	P5
Variables	Clearly define all outcomes, exposures, predictors, potential confounders, and effect modifiers. Give diagnostic criteria, if applicable	P5-6
Data sources/ measurement	* For each variable of interest, give sources of data and details of methods of assessment (measurement). Describe comparability of assessment methods if there is more than one group	P5-6
Bias	Describe any efforts to address potential sources of bias	P7
Study size	Explain how the study size was arrived at	P7
Quantitative variables	Explain how quantitative variables were handled in the analyses. If applicable, describe which groupings were chosen and why	P5-6

Statistical methods	(a) Describe all statistical methods, including those used to control for confounding	P6-7
	(b) Describe any methods used to examine subgroups and interactions	P6-7
	(c) Explain how missing data were addressed	P7
	(d) If applicable, describe analytical methods taking account of sampling strategy	NA
	(e) Describe any sensitivity analyses	P7
Results		
Participants	(a) Report numbers of individuals at each stage of study—eg numbers potentially eligible, examined for eligibility, confirmed eligible, included in the study, completing follow-up, and analysed	P7
	(b) Give reasons for non-participation at each stage	P7
	(c) Consider use of a flow diagram	NA
Descriptive data	(a) Give characteristics of study participants (eg demographic, clinical, social) and information on exposures and potential confounders	P7
	(b) Indicate number of participants with missing data for each variable of interest	P7
Outcome data	Report numbers of outcome events or summary measures	P7 & Table 1
Main results	(a) Give unadjusted estimates and, if applicable, confounder-adjusted estimates and their precision (eg, 95% confidence interval). Make clear which confounders were adjusted for and why they were included	P7-8, Tables 2-4
	(b) Report category boundaries when continuous variables were categorized	NA
	(c) If relevant, consider translating estimates of relative risk into absolute risk for a meaningful time period	NA
Other analyses	Report other analyses done—eg analyses of subgroups and interactions, and sensitivity analyses	P8
Discussion		
Key results	Summarise key results with reference to study objectives	P8-9

Limitations	Discuss limitations of the study, taking into account sources of potential bias or imprecision. Discuss both direction and magnitude of any potential bias	P10
Interpretation	Give a cautious overall interpretation of results considering objectives, limitations, multiplicity of analyses, results from similar studies, and other relevant evidence	P8-10
Generalisability	Discuss the generalisability (external validity) of the study results	P10 where we stated that our analysis was based on a nationally representative sample
Other information		
Funding	Give the source of funding and the role of the funders for the present study and, if applicable, for the original study on which the present article is based	P3 – Acknowledgement